# Effects of COVID-19 Restriction Policies on Urban Heat Islands in Some European Cities: Berlin, London, Paris, Madrid, and Frankfurt

**DOI:** 10.3390/ijerph19116579

**Published:** 2022-05-28

**Authors:** Taher Safarrad, Mostafa Ghadami, Andreas Dittmann

**Affiliations:** 1Geography and Urban Planning Department, University of Mazandaran, Babolsar 13534-47416, Iran; t.safarrad@umz.ac.ir; 2Department of Geography, Justus Liebig University Giessen, 35390 Giessen, Germany; andreas.dittmann@geogr.uni-giessen.de

**Keywords:** COVID-19, urban heat island, restriction policies, European cities

## Abstract

The present study investigates the effects of policies restricting human activities during the COVID-19 epidemic on the characteristics of Night Land Surface Temperature (NLST) and Night Urban Heat Islands (NUHI) in five major European cities. In fact, the focus of this study was to explore the role of anthropogenic factors in the formation and intensity of NUHI. The effect of such factors was uncontrollable before the COVID-19 outbreak on the global scale and in a real non-laboratory environment. In this study, two indices, the concentration of Nitrogen dioxide (NO_2_) and Nighttime Lights (NL), were used as indicators of the number of anthropogenic activities. The data were collected before the COVID-19 outbreak and after its prevalence in 2019–2020. A Paired samples *t*-test and a Pearson correlation were used to examine the differences or significant relationships between the variables and indicators studied throughout the two periods. The results of the study confirmed a direct and significant relationship between NO_2_ and NL indices and the NUHI and NLST variables; however, using strict restrictions during the COVID-19 pandemic, the NO_2_ and NL indices decreased seriously, leading to significant changes in the characteristics of the NUHI and NLST in the five cities. This study has some implications for urban planners and politicians, e.g., the environmental impacts of changing the nature and level of anthropogenic activities can greatly affect the pattern and intensity of the Urban Heat Islands (UHIs) (as a serious environmental challenge).

## 1. Introduction

The sixth IPCC report shows that the average global temperature has alarmingly risen during the last four decades. The global surface temperature in the first two decades of the 21st century (2001–2020) was 0.99 °C higher than 1850–1900. It was 1.09 °C higher in 2011–2020 than 1850–1900, with larger increases overland (1.59 °C) than over the ocean (0.88 °C) [1]. At the same time, the results of various studies show that compared to the general trend of global warming, cities, and, most specifically, densely populated cities or metropolises, have had a faster warming trend [2,3,4], so that urban areas have experienced 2–3 °C increase in temperature during the 20th century [5,6]. This indicates that urban spaces are getting warm at a much higher speed under the influence of general atmospheric warming as well as the phenomenon of UHI. The importance of this issue will be exacerbated by the fact that by 2050, about 70% of the world’s population is projected to live in cities [7,8,9,10,11]. The phenomenon of UHI has become a serious threat to urban life in the last few decades due to the anomalous physical expansion of urban areas [12,13,14]. In this regard, the issue of extreme temperature has become one of the deadliest natural disasters in Europe. From 1980 to 2017, heat waves have caused 68% of the natural-hazard-induced deaths and 5% of damage to the European economy [15]. 

Given the importance and effects of the phenomenon of UHI on the health and quality of urban life, it has been investigated by several researchers in different fields following its first introduction by Howard in 1833 [16] and after its use by Oke [17,18,19,20,21] and Chandler [22]. UHI is the most prominent feature of urban climate, especially in densely populated cities [5,23]. It actually refers to the urban areas that are warmer than the rural or natural environment around them [19]. UHIs in urban areas are formed due to extensive changes in the nature and materials of the land surface and can expand and intensify. One of the first effects of the spatial expansion of cities is a change in land use and coverage [24]. Changing the vegetation coverage and barren lands to built-up areas can cause significant changes in the energy balance and, consequently, the temperature of the city. Developing cities on natural lands causes the expansion and replacement of impervious urban surfaces that retain less moisture compared to the natural coverages. Hence, the significant differences in heat capacity, roughness, and Albedo collectively make the urban lands warmer than their surrounding rural environments [14,25,26,27,28,29]. The effects of UHI can be intensified by heat emitted from transportation, heating systems in homes, service centers, and industrial environments, as well as indoor ventilation systems [19].

Most research conducted so far on the UHI has examined the factors affecting their formation and their effects on urban health and environmental quality [30,31,32,33,34,35,36]. They have also explored the relationship between the formation and intensity of UHI and the set of urban physical variables in detail [37,38,39]. In this regard, physical variables, such as physical characteristics of the land surface, types of urban land use, urban density, the spatial pattern of building blocks, as well as the status of mixed land uses, have been studied [40,41,42,43,44,45,46]. In addition, some other researchers have investigated the relationship between the demographic and economic variables of cities (e.g., population density, economic characteristics, and traffic congestion) and the formation, expansion, and intensity of UHI. The following four factors are the main reasons for the creation of a UHI [5]: Increased anthropogenic heat release
Heat release resulting from energy consumption in urban areasChanges in surface cover
Reduced surface evapotranspiration capacity due to less green areaThe heat storage effect of construction materials such as concrete and asphaltUrban structure
Heat stagnation due to densely packed buildingsExpansion of urban areasOther
The greenhouse effects of fine-particulate air pollution in the urban atmosphere

The clear point in the previous research is that they are less researched compared to the physical properties of the land surface, despite the importance of anthropogenic factors in the development and expansion of the UHI. This may be due to limitations in the control of anthropogenic factors in determining their contribution and role in the formation and expansion of UHIs in an objective and large-scale non-laboratory conditions [47]. The effect of physical factors of land cover on the UHI can be measured with high accuracy using a variety of objective data on an urban scale. However, controlling and separating the share of anthropogenic factors in this regard is difficult and impractical in an objective environment. This gap in the relevant literature has driven us to compile this research to investigate the impact of anthropogenic factors on LST changes and UHIs during the lockdown. 

The outbreak of the COVID-19 pandemic in late 2019 led to the imposition of unprecedented strict measures on human activities in different countries of the world [48]. During this period, a diverse range of restricting policies, including complete lockdown, distance working, travel restrictions, and temporary closure of many commercial and industrial businesses, were applied [49,50,51,52]. These mentioned measures have had significant effects on environmental variables on a local and global scale [53,54,55]. For example, the data show that global greenhouse gas emissions in early April 2020 were decreased by about 17% compared to the average of 2019 [56]. The results show that the rate of greenhouse gas emissions during the global lockdown period decreased significantly and unprecedentedly since World War II [57]. Data also show a 50% decrease in inland transportation and a 35% decrease in industrial activities during the lockdown [56].

Severe restrictions, especially during the complete lockdown, provided good conditions for researchers to control and measure the effect of anthropogenic factors on the formation and expansion of UHIs. In fact, during the COVID-19 epidemic, it was possible to study the contribution of the anthropogenic factor to the formation of the UHI separately from other physical factors [53,58,59,60]. Given these conditions and the gap in the literature on the impact of anthropogenic factors on land surface temperature and the intensity of UHIs, this study intends to use satellite images and remote sensing models to determine the impact of lockdown measures during the COVID-19 era on the pattern and intensity of UHIs and Land Surface Temperature (LST) in some of the most populous cities in Western Europe. 

## 2. Study Settings

The study area includes five major and important cities in Europe in which significant restrictive policies have been implemented during the COVID-19 epidemic: London, Berlin, Frankfurt, Paris, and Madrid (Figure 1). As mentioned earlier, when the World Health Organization publicly declared COVID-19 as a global epidemic, all countries were required to implement a variety of health policies. Given that the authorities of the cities studied in this research have not simultaneously adopted the restricting policies, to increase the validity of the results, a common time frame among all the samples studied has been selected, during which all the cities were in a state of complete lockdown. In this study, the analysis of maps and the average of the studied variables was done from 23 March to 20 April, which exactly coincided with the period of executing the most severe restrictions in the studied cities. Thus, the years 2019 and 2020 represent the above-mentioned time period (Figure 2).

## 3. Materials and Methods

### 3.1. Data

#### 3.1.1. Nighttime Urban Heat Island (NUHI) and Nighttime Land Surface Temperature (NLST)

To investigate the spatial characteristics of the LST in the studied cities and their non-built up surrounding areas, MOD11A2 version 6 data were used. MOD11A2 version 6 provided the average Land Surface Temperature and Emission (LST & E) during 8 days (10:30 a.m. and 10:30 p.m.) with 1 km spatial resolution in a 1200 × 1200 km grid. Each pixel value in the MOD11A2 is a simple average of all the corresponding MOD11A1 LST pixels collected within that eight-day period [61].

#### 3.1.2. UHI Intensity

Urban impervious surfaces (e.g., roads, buildings, and other man-made structures) have a higher heat capacity than pervious surfaces (e.g., vegetation and barren lands). Accordingly, the temperature difference between the urban area and surrounding areas often increases on calm and cloudless days and in the last hours of the day. More precisely, the maximum intensity of the UHIs mostly occurs three to five hours after sunset [62,63,64]. Given that, in this research, to investigate the status of the NUHI in the study samples, MOD11A2 nighttime images (10:30 p.m.) have been used. Moreover, to calculate the intensity of the NUHI, first, the land cover layer was prepared using CGLS-LC100 images. A dynamic land cover map at 100 m resolution (CGLS-LC100) is a new product of the CGLS that provides a global land cover map at 100 m spatial resolution. It is derived from the PROBA-V 100 m time series [65].

Using the above-mentioned images, the urban areas were separated from the surrounding area. For this purpose, the Built-Up codes in the CORINE Land Cover (CLC) images, corresponding to the physical boundaries of the cities, were considered as urban areas. Then, all areas with natural land covers (pervious surfaces such as barren land, green space, and agricultural land) and with a distance more than 3 km from the cities were separated as a non-built up surrounding area. Due to the impact of the cities on their surroundings, the temperature data related to the first 3 km distance from the city and its surrounding area were not included in the analysis. The latest CORINE Land Cover (CLC) data update for 2019 was used to detect the pervious surfaces (Figure 3).

After determining the urban areas and their surrounding area, the NUHI was calculated with the following formula: NUHI = NLST − NLSTr(1)

NLST is the nighttime land surface temperature of the city and NLSTr is the nighttime land surface temperature of the surrounding area generalized to the urban areas. 

After determining the urban area and its surrounding areas, as mentioned earlier, first, the NLST of each point of the city and its surrounding areas was calculated using MOD11A1 images. With the Ordinary Kriging method for each urban point (cell), the NLST value of its nearest point (cell) from the surrounding areas was also estimated (RMSE: 0.43). Finally, two NLSTs were calculated for each cell within the city area (the NLST of a point inside the city and the NLST of a point in the surrounding area). Then, the difference between these two values was calculated according to equation 1 as the NUHI of the urban area. After calculating the NLST and NLSTr in each city, the NUHI values were calculated. This method gives a more detailed and accurate understanding of the pattern of the UHIs compared to the methods that use a gross average number for the city and its non-built up surrounding area.

#### 3.1.3. Nitrogen Dioxide (NO_2_)

Nitrogen dioxide (NO_2_) and nitrogen monoxide (NO) are commonly known as nitrogen oxides (NO_x_ = NO + NO_2_). They are important trace gases in the Earth’s atmosphere, present in both the troposphere and the stratosphere (notably fossil fuel combustion and biomass burning) and natural processes (such as microbiological processes in soils, wildfires, and lightning). In the present study, the spatial pattern of NO_2_ concentration has been used as a variable representing the temporal and spatial pattern of anthropogenic activities in the two periods before the outbreak of the COVID-19 virus and after the execution of strict lockdown regulations. These data show the total vertical column of NO_2_ (ratio of the slant column intensity of NO_2_ and the total air mass factor) in mol/m^2^ with a pixel size of 1113.2 m.

#### 3.1.4. Nighttime Lights

The Earth Observation Group (EOG) is an expert at working with overnight remote sensing data. The introduction of the JPSS series of satellites with advanced DNB sensors has provided more possibilities in the field of remote sensing. The Day Night Band (DNB) on the VIIRS sensor is a panchromatic band that can detect very dim night scenes. EOG is used to produce and supply annual Nighttime Light (NL), while recently, it has started monthly production of these products as well [66]. In this study, the monthly data obtained from VIIRS Stray Light Corrected Night time Day/Night Band Composites Version 1 have been used to monitor the NLs in the studied cities. These images show the average DNB radiance values in Nano Watts/cm^2^/sr with a pixel size of 463.83 m.

### 3.2. Analysis Method

It should be initially pointed out that in this study, all the relevant data, information, and analysis belonged to two main time periods, namely 2019 (before the onset of the COVID-19 epidemic) and 2020 (after the onset of the COVID-19 epidemic and the application of restricting regulations on anthropogenic activities). Furthermore, for selecting a specific time period to measure and evaluate the effects of the COVID-19, a time period has been chosen during which the cities studied have experienced a similar situation.

This research consists of three main steps (Figure 4). In the first step, the main data required for the research were collected through four types of satellite images described in the previous section. In the second step, the information required for the research was extracted from the mentioned data, i.e., the NLST and NLSTr data were extracted from MOD11A2 images, urban and surrounding area data were extracted from CGLS-LC100 images, the total vertical column of NO_2_ data was extracted from Copernicus Sentinel-5P images, and the average DNB radiance data were extracted from VIIRS images. In the third step, a paired-samples *t*-test and correlation analysis were used to investigate the possible effects of lockdown periods on the dependent variables. For this purpose, a raster layer with a resolution of 1000 × 1000 m (as shown in Figure 3) was produced for each of the main variables of the research separately for 2019 and 2020. The cell values of each new raster layer were calculated based on the values of the nearest pixel of the image related to the studied variable. At this stage, each of the research variables, including the NLST, NUHI, NO_2_, and NL, had a raster layer with the same resolution in the two time periods. Accordingly, two raster layers from each of the variables with two different time periods helped to make accurate statistical comparisons between the variables. A paired-samples *t*-test was used to examine significant changes in the characteristics of research variables before the COVID-19 outbreak (2019) and after the application of restricting policies (2020). A Pearson correlation test was also used to investigate the relationship between the NUHI changes and the NLST, NO_2_, and NL changes.

## 4. Result

### 4.1. Difference in Averages NLST in 2019 and 2020

An examination of the average NLST from 23 March to 20 April in 2019 and 2020 shows that, in general, the centers of the studied cities were warmer than their surrounding area during both periods. At the same time, the average urban NLST showed a significant difference between the studied cities. The average NLSTs of Berlin, Frankfurt, and Madrid in 2020 showed lower values (getting cooler) compared to in 2019, while the NLST averages of London and Paris showed that these two cities got warmer in 2020 compared to 2019 (Figure 5 and Figure 6, Table 1).

The higher temperature of the NLST in Paris and London from 23 March to 20 April 2020, seems to be due to the higher temperature of these regions as compared to the previous year (2019), being irrelevant to anthropogenic activities and COVID-19 restrictions. To confirm this, the NLST averages of 23 March to 20 April have been shown for the years 2019 and 2020 in Figure 7. As it can be observed, the NLST difference was positive in Paris and London, as well as their surrounding areas, which can be a result of changes in global circulation patterns in 2020 in comparison to 2019.

### 4.2. Difference in Average NUHI in 2019 and 2020

The results show that in contrast to the NLST variable, London and Paris cities experienced a significant decrease in the NUHI in 2020 compared to 2019 (Figure 8). However, in contrast, this variable significantly increased in Berlin (Figure 9 and Table 2). As London and Paris cities experienced an increase in the NLST in 2020, these results were somewhat weird and higher NUHI values were expected for these cities in 2020. In addition, for Berlin, due to the decrease in the NLST, a proportional decrease in the NUHI values was expected. As the findings of this study indicated, there has been a strong and significant relationship between NLST and NUHI values in the studied cities (Figure 10 and Table 3).

To have a more accurate evaluation, the spatial characteristics of the NUHI were studied based on the changes in the areas of the NUHI zones in the studied cases. For this purpose, the NUHI were divided into two classes: Zones with a temperature above 2 °C and zones with a temperature below 2 °C (Table 4). The results showed that during 2020 and simultaneous with the implementation of lockdown and policies restricting anthropogenic activities, in the cities of London, Paris, and Madrid, the area of zones with NUHI above 2 °C decreased by 152, 43, and 44 km^2^, respectively. Therefore, a change in the area of the NUHI zones has been observed in these cities besides a decrease in the NUHI. In the case of Berlin, however, this change has been the reverse, i.e., during the lockdown, in addition to the increase of NUHI in Berlin, the area of zones with NUHI more than 2 °C increased by about 75 km^2^.

As shown previously, despite the strong and significant relationship between the NLST and NUHI, different results have been observed regarding the rate of change in these two variables between the studied cities. To justify such a finding, anthropogenic activities in the studied cities were investigated. Since COVID-19 restrictions have had the greatest impact on anthropogenic activities and urban life (especially nightlife), to track these changes, the two variables of NO_2_ concentration and NL were examined, as they were directly related to urban activities and nightlife in the cities. 

The results show that the restrictions applied against COVID-19 significantly affected the values of NO_2_ and NL variables (Figure 11, Figure 12, Figure 13 and Figure 14, as well as Table 5 and Table 6). Policies restricting anthropogenic activities during the COVID-19 outbreak have significantly reduced the urban pollution concentration (NO_2_) and the NLs in the studied cities. However, it should be noted that the rate of change in these two variables, depending on the seriousness and strictness of local authorities in applying the restricting regulations and public participation in the studied samples have been different. For example, among the studied cities, the values of NO_2_ and NL decreased the least in Berlin and the most in London in 2020. This result can justify the reasons for the difference in the rate of the NUHI between the studied cities in 2020, especially for Berlin (Figure 11, Figure 12, Figure 13 and Figure 14).

In the following, the relationship between the changes in NO_2_ concentration and NLs and the changes in the NUHI was investigated in the two time periods of the study. For this purpose, first, the possibility of significant changes in the NLs and NO_2_ concentration variables in 2019 and 2020 was investigated in the sample cities (Table 7), and then, the relationship between the changes in NO_2_ concentration and NLs and the changes in the NUHI was investigated using a Pearson correlation test (Table 8).

In the next step, the relationship between the changes in NUHI and each of NLST, NL, and NO_2_ variables was investigated in the studied cities. The results showed that there is a direct and significant relationship between the NUHI changes and changes in these variables (Figure 15). To increase the validity of the findings, the number of cities studied was increased. Therefore, in addition to the previously mentioned five cities, 10 other European cities (the total number of samples increased to 15, including the previous five ones) were selected in the same time frame, when strict COVID-19 restrictions were imposed (Figure 16). Then, the statistical correlation tests were used to investigate the possibility of a significant relationship between the above-mentioned variables (Table 8 and Table 9).

The results showed that the NUHI changes have a significant direct relationship with NO_2_ concentration changes. Accordingly, increasing NO_2_ concentration leads to an increase in the NUHI (Figure 17). As mentioned earlier, the main purpose of this study is to investigate the impact of policies restricting anthropogenic activities during the COVID-19 outbreak on the UHI. Thus, confirming the significant and direct relationship between NO_2_ concentration (as an important variable indicating anthropogenic activities) and the NUHI changes can well show the role of anthropogenic factors in this field. This was impossible before the rigid lockdowns of COVID-19. 

## 5. Discussion

The findings of this study showed that physical factors alongside anthropogenic factors, e.g., economic and social activities, can have a profound effect on the formation and intensification of NUHIs. As mentioned earlier, numerous studies have been conducted on the impact of physical factors of land cover on land energy balance and urban temperature changes; however, it was not possible to control and isolate the impact of anthropogenic factors before applying strict COVID-19 lockdown policies in real non-laboratory conditions. The results of this study showed that the conditions caused by the COVID-19 epidemic had a significant effect on the NLST characteristics and the NUHI pattern in the studied European cities. However, the difference between the average NLST of 2019 and 2020 was not the same between the studied cities. While some cities got cooler, London and Paris showed an increase in the average NLST. Therefore, we have not seen a similar study examining the NLST changes during the COVID-19, but research findings [67] in the Indo-Gangetic Basin show that Daytime LST decreased during the lockdown. It should be noted, however, that this decrease was more in rural areas than in urban areas. Since the gross mean values did not show all the details of LST changes in the cities, in this study, the pattern of the NUHI was also examined. The findings show that Paris, London, and Madrid cities experienced a decrease in the NUHI area (above 2 °C) during the study period, while Berlin showed an increase in the NUHI area. This finding is in contrast with those obtained from the other studied cities. At the same time, the correlation between the NUHI and NLST in both periods was direct and positive for each of the studied cities. It should be noted that the values related to the NUHI and NLST variables and other variables were calculated separately for pixels and entered into the statistical tests. Differences in the NUHI and NLST values indicate that the studied cities did not respond equally to the lockdown conditions. In fact, the application of strict lockdowns never experienced before has been difficult for many people and even for city officials. Consistent with the results of this study, ref. [68] showed in their study on the effects of COVID-19 lockdown on UHIs in Pakistan that from 23 March to 30 April, the mean decrease in Surface Urban Heat Island (SUHI) in large cities was about 20% or 0.4 °C. However, in most important cities of Pakistan, the SUHI decrease was nearly 0.7 °C. Regarding the impact of human behaviors on heat islands, research by [69] was conducted on the impact of COVID-19-related traffic slowdown on UHI characteristics. This study showed that an 80% reduction in road traffic can reduce the near land surface temperature by more than 1 °C in areas with heavy traffic. Since this result is only true around traffic corridors, the impact of non-traffic factors on increasing the LST should be examined in farther areas. In this regard, the application of transportation restrictions has severely reduced the SUHI effect in large cities of Pakistan [68]. The significant impact of traffic flow on the UHI phenomenon has also been confirmed in research [70]. Contrary to the results of this study, research [67,71,72] shows that the UHI intensity has increased during the lockdown days. In this regard, one of the factors affecting the increase in UHI intensity during the lockdown is the reduction of air pollutants. Based on the findings of the mentioned research, the reduction of atmospheric pollutants has increased the incoming surface radiation and, consequently, has increased the LST and the UHI intensity in the studied samples [67]. Moreover, the difference in the amount of air pollutants between the urban and rural areas is an effective factor in the UHI intensity [71,72]. Due to the fact that the UHI intensity is calculated based on the temperature difference between the city and its surrounding rural or natural areas, the restrictions during the lockdown in the process of agricultural activities can also be studied in this regard. According to the findings of [67,73], the lockdown policies in the agricultural sector have caused a delay in the harvest time of agricultural products, leading to the presence of vegetation coverage for a longer period of time, which in turn, has been a factor in modulating the LST in agricultural areas and increasing SUHI. The important point of this research is that the lockdown during the COVID-19 delayed the harvest of agricultural products in rural areas. Therefore, the temperature difference between the urban and rural areas increased, and this had a direct effect on the intensity of the daytime SUHI, i.e., the cooling of rural areas (rather than the warming of urban areas) has increased the intensity of the daytime SUHI.

In this study, the changes in NO_2_ concentration and NLs were used to investigate the effects of lockdown on anthropogenic activities. The relevant results showed that NO_2_ concentration can be an indicator to show the anthropogenic activities in cities. The comparison of NO_2_ concentration in 2019 and 2020 showed that with the decrease of anthropogenic activities in the cities, the NO_2_ concentration decreased sharply. This was also the case with the NLs. The NO_2_ concentration can well indicate the seriousness and success of local and central governments in implementing policies to limit the spread of the COVID-19 epidemic. The results of the present study are consistent with [54] in New York city and with [68] in major cities in Pakistan, showing that the restrictions on the anthropogenic activities during the COVID-19 reduced NO_2_ emissions by 40% in coal-based power plants and by 30% in main urban areas. During the COVID-19 pandemic, the decrease of the anthropogenic activities could reduce air pollutants by about 5–6% and, in particular, caused a 14% reduction in NO_2_ and 19% reduction in SO_2_ [67]. This issue was also confirmed in [74]. Simultaneous with the lockdown policies in India, a reduction of more than 45% was reported in atmospheric aerosols in some Indian provinces [75]. Furthermore, compared to the previous year, the release of NO_2_, SO_2,_ and PM_2.5_ decreased in the Yangtze River Delta region in China [76]. A decrease in the concentration levels of NO_2_ and PM_2.5_ was reported in most cities of China [77,78,79,80] and other parts of the world [81,82,83,84,85]. Furthermore a significant decrease was observed in CO, NO_2,_ and PM_2.5_ levels (49%, 35%, and 21%, respectively) in Almaty and Kazakhstan [81]. Similarly, it was shown in [60] that during the lockdown period, due to the reduced effects of anthropogenic activities, the air quality of many European countries improved. For example, ref. [85] shows that the concentration of NO_2_ during these days decreased between 20–30% in France, Spain, Italy, and Germany.

Based on the findings of this study, in terms of NO_2_ concentration in 2020, it can be said that among the cities studied, London has been more successful in enforcing the COVID-19 restrictions, while Berlin has been weaker than the other cities under study. In addition, the main results of the study showed that there is a direct and significant relationship between the NUHI changes and NO_2_ concentration changes. This information indicates the serious impact of anthropogenic activities on the formation of the UHIs. This research has contributed to the urban climate literature in some ways, e.g., it has used satellite images to investigate the relationship between the NUHI and the NLST, NO_2_ concentration, and NL variables with respect to the impact of the COVID-19 lockdown. The restrictions on the anthropogenic activities during the COVID-19 period were considered a special opportunity for the research team to examine the role of the anthropogenic factors in LST changes and UHIs. In this regard, due to the conditions caused by the COVID-19 pandemic, inadequate research has focused on the urban climate literature. Another positive aspect of this research is to study 15 European cities (five main cities and 10 control cities), which has increased the generalizability of the research and the validity of its findings.

## 6. Conclusions

The main purpose of this study was to investigate the effects of the COVID-19 restriction measures on the NLST and NUHI changes. For this purpose, by examining the five main European cities and adding 10 other control European cities (during the research), the effects of policies restricting the anthropogenic activities during the COVID-19 on the NLST and NUHI characteristics were investigated. In other words, the focus of this study was to investigate the role of anthropogenic factors in the NLST and NUHI characteristics. Notably, before the outbreak of the COVID-19 pandemic, it was basically impossible to control anthropogenic activities on a global scale. Therefore, the lockdown conditions were considered a serious opportunity for the research team to test this issue. By comparing the data belonging to the two time periods before COVID-19 and after the COVID-19 virus outbreak in 2019–2020, the research results showed that the anthropogenic factors in a set of daily activities, e.g., communication, as well as industrial, service, and administrative activities, have had a significant impact on the formation and pattern of the NUHI and NLST changes. However, the result of LST was not the same between the studied cities. While some cities showed cooler LST, London and Paris recorded an increase in the average NLST. On the other hand, the result of NUHI showed that Paris, London, and Madrid cities experienced a decrease in the NUHI area (above 2 °C) during the study period, while Berlin showed an increase in the NUHI area. Differences in the NUHI and NLST values indicate that the studied cities did not respond equally to the lockdown conditions. In fact, the application of strict lockdowns, never experienced before, has been difficult for many people and even city officials.

In this study, two indices of NO_2_ concentration and NL were used to measure the anthropogenic activities; both of which, especially the index of NO_2_ concentration, showed well the results of implementing the restricting policies during the COVID-19 pandemic. Thus, with the beginning of the implementation of rigid restrictions, the NO_2_ and NL concentration indices decreased significantly. Given the confirmed significant relationship between the NO_2_ and NL concentration indices and the number of anthropogenic activities, in the continuation of the research, the relationship between these indices and the NLST and NUHI characteristics was examined. At this stage, in addition to the five main cities, including London, Paris, Berlin, Frankfurt, and Madrid, 10 other European cities were included in the statistical analysis to increase the validity and generalizability of the results. The results of the correlation test confirmed a direct and significant relationship between the NO_2_ and NL concentration indices and the NLST and NUHI characteristics. This result showed that the anthropogenic factors had a significant and serious impact on the NLST and NUHI characteristics so that the extent and intensity of the NUHI would be greatly reduced by limiting the anthropogenic activities or implementing environment-friendly policies in such activities.

This research had limitations such as, first, the lengthy data collection process from the European cities, and examining the special regulations of the COVID-19 period took a great deal of research time. Second, collecting data was in the native language of the countries, and translating data was time-consuming. Third, due to a lack of financial resources, developing the research scope and simultaneously studying the impact of physical and anthropogenic factors together in an integrated model were not possible. Accordingly, due to the specific circumstances caused by COVID-19 restrictions, future researchers are suggested to consider the contribution and effect of the anthropogenic and physical land cover factors on the formation and spatial pattern of the UHIs. The findings of this study are important for urban planners and policymakers, implying that by managing the anthropogenic activities, especially in the field of urban transportation and heating systems of industrial, commercial, and residential units, the intensity and extent of the UHIs can be effectively controlled (as a serious challenge to sustainable urban development).

## Figures and Tables

**Figure 1 ijerph-19-06579-f001:**
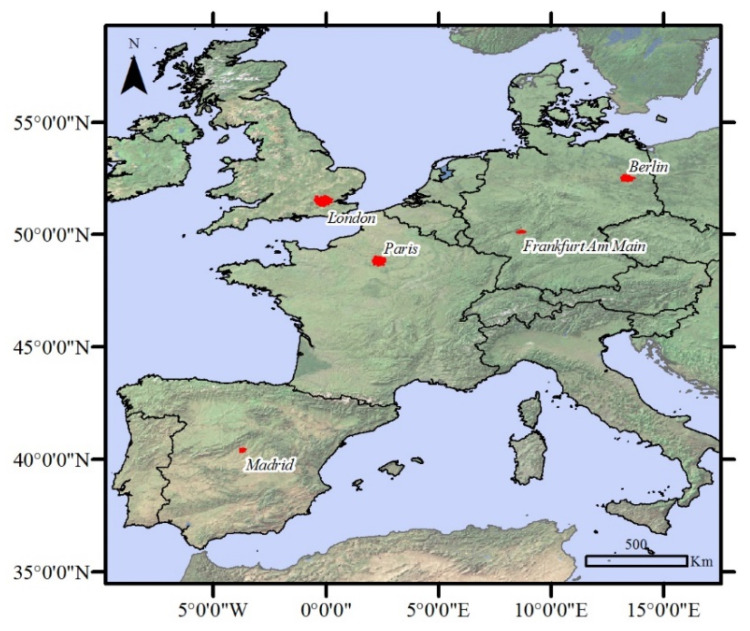
Location of study cases.

**Figure 2 ijerph-19-06579-f002:**
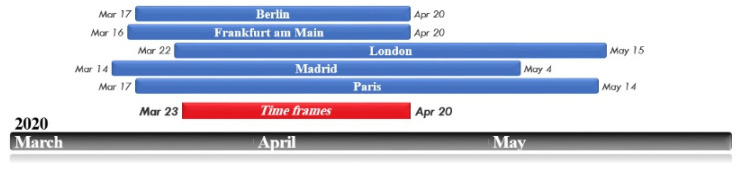
Time frames for severe restrictions on COVID-19 in each city and a common time frame for the whole study area.

**Figure 3 ijerph-19-06579-f003:**
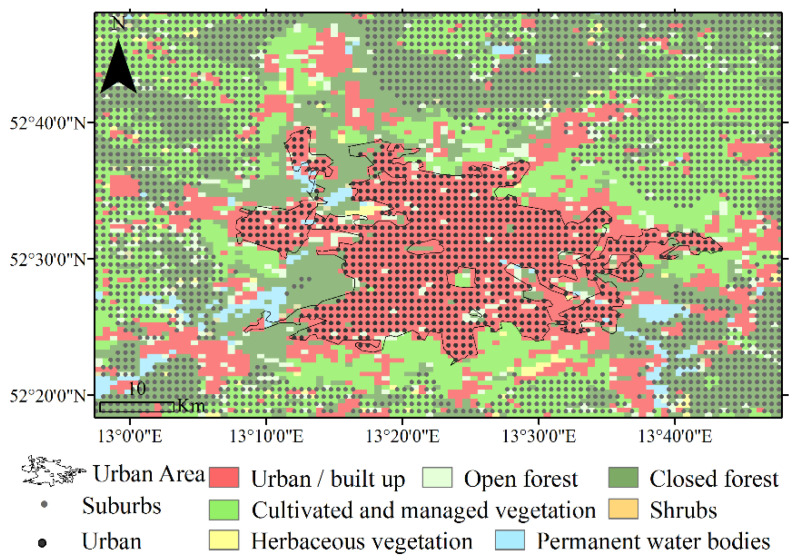
A sample land cover image to separate the city of Berlin from its surroundings (similar images were used for the other cities under study).

**Figure 4 ijerph-19-06579-f004:**
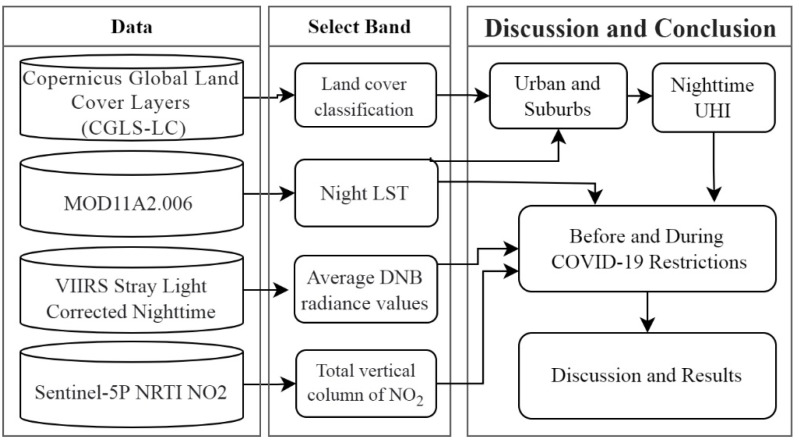
Research process.

**Figure 5 ijerph-19-06579-f005:**
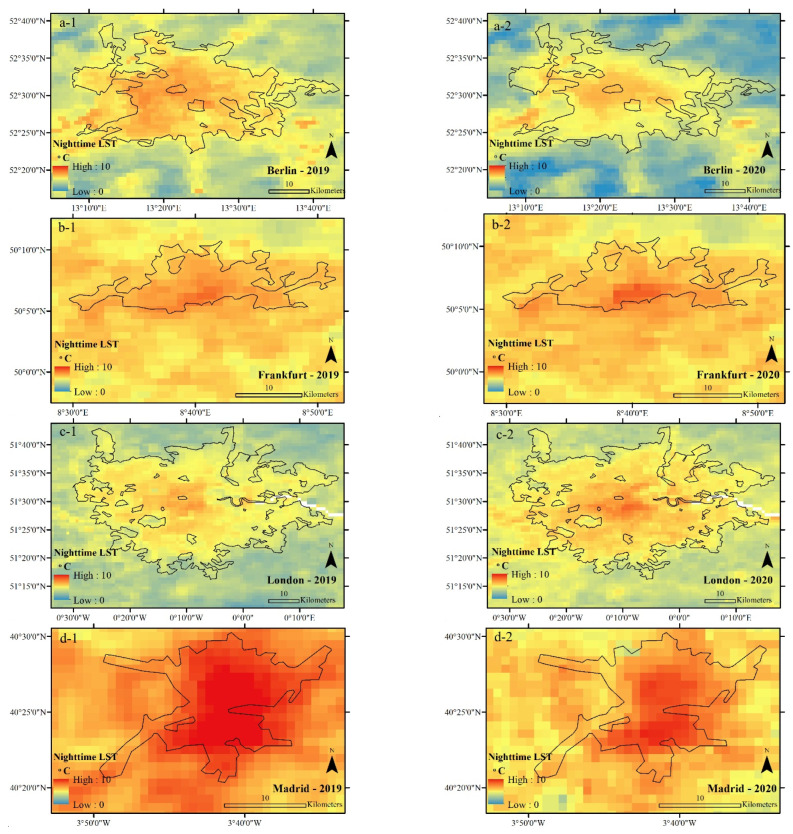
Average NLST of the studied cities during CSR, extracted from MOD11A2 data in °C. (**a-1**) Average NLST of Berlin in 2019, (**a-2**) Average NLST of Berlin in 2020, (**b-1**) Average NLST of Frankfurt in 2019, (**b-2**) Average NLST of Frankfurt in 2020, (**c-1**) Average NLST of London in 2019, (**c-2**) Average NLST of London in 2020, (**d-1**) Average NLST of Madrid in 2019, (**d-2**) Average NLST of Madrid in 2020, (**e-1**) Average NLST of Paris in 2019, (**e-2**) Average NLST of Paris in 2020.

**Figure 6 ijerph-19-06579-f006:**
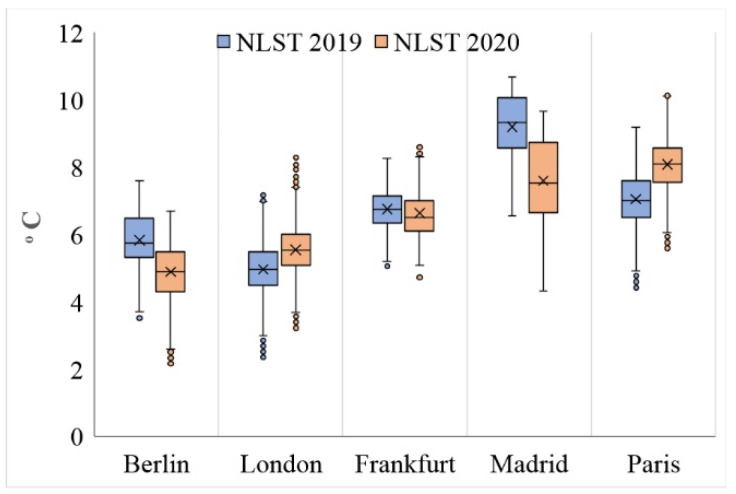
NLST characteristics in the studied cities during 2019 (blue) and 2020 (red).

**Figure 7 ijerph-19-06579-f007:**
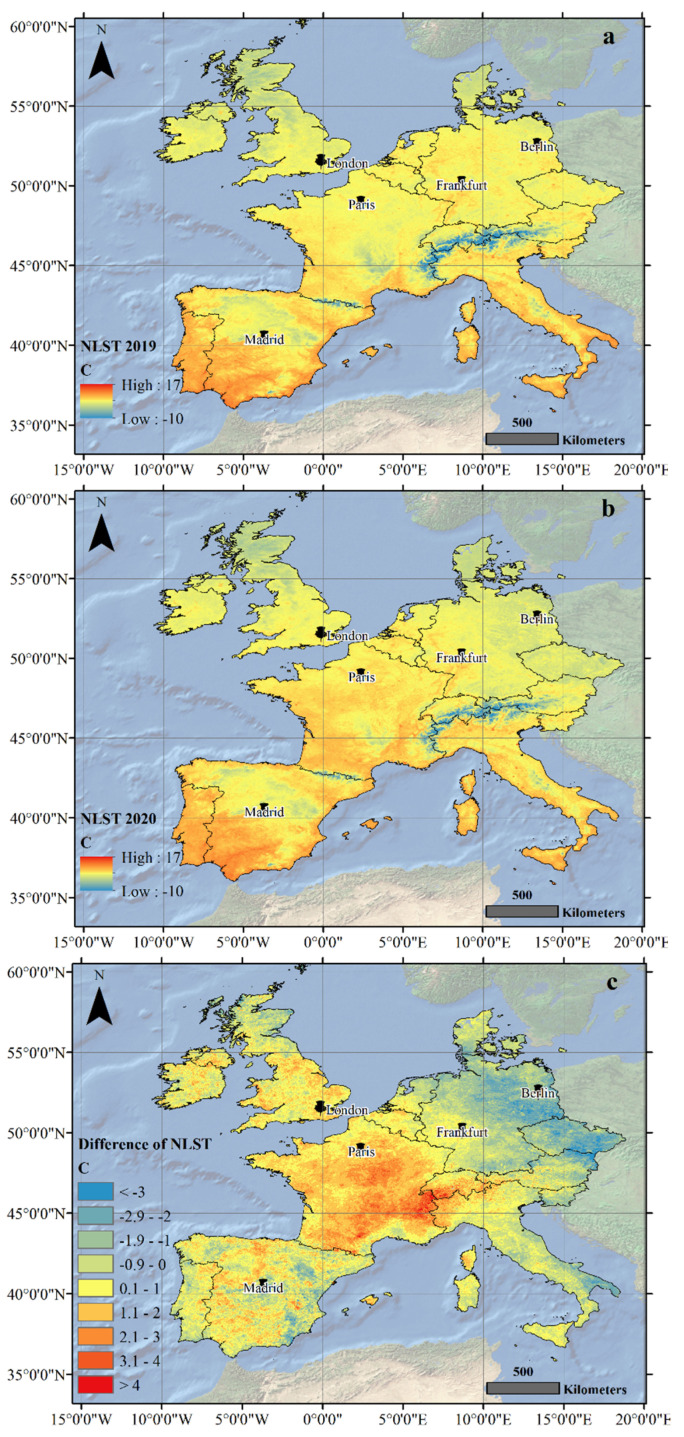
The average NLST of the study area from 23 March to 20 April in (**a**) 2019 and (**b**) 2020 and (**c**) the difference between the two years investigated, obtained from MOD11A1 images in centigrade degrees.

**Figure 8 ijerph-19-06579-f008:**
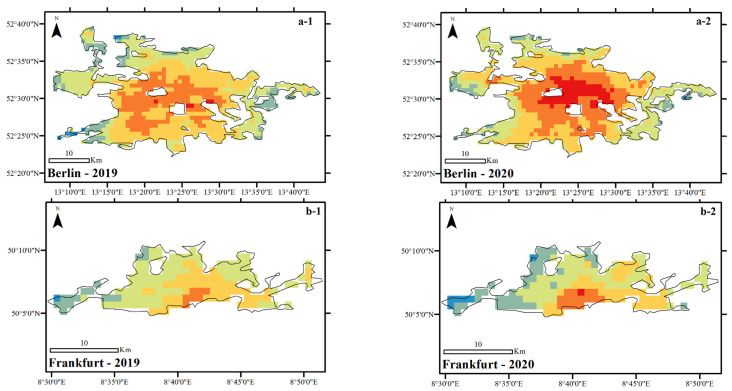
Average NUHI of the studied cities during CSR, extracted from MOD11A2 data in °C. (**a-1**) Average NUHI of Berlin in 2019, (**a-2**) Average NUHI of Berlin in 2020, (**b-1**) Average NUHI of Frankfurt in 2019, (**b-2**) Average NUHI of Frankfurt in 2020, (**c-1**) Average NUHI of London in 2019, (**c-2**) Average NUHI of London in 2020, (**d-1**) Average NUHI of Madrid in 2019, (**d-2**) Average NUHI of Madrid in 2020, (**e-1**) Average NUHI of Paris in 2019, (**e-2**) Average NUHI of Paris in 2020.

**Figure 9 ijerph-19-06579-f009:**
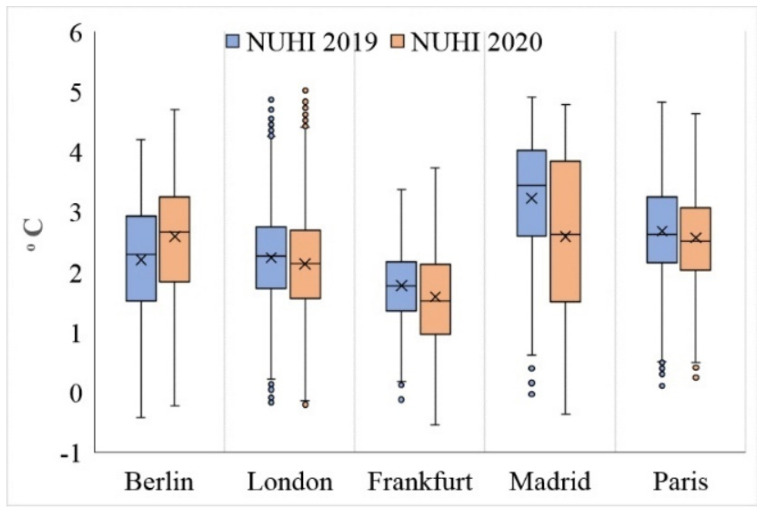
NUHI characteristics in the studied cities during 2019 (blue) and 2020 (red).

**Figure 10 ijerph-19-06579-f010:**
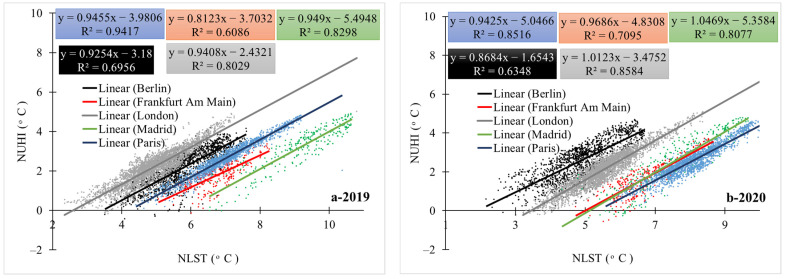
(**a-2019**) relationship between NLST and NUHI in the studied cities 2019. (**b-2020**) relationship between NLST and NUHI in the studied cities 2020.

**Figure 11 ijerph-19-06579-f011:**
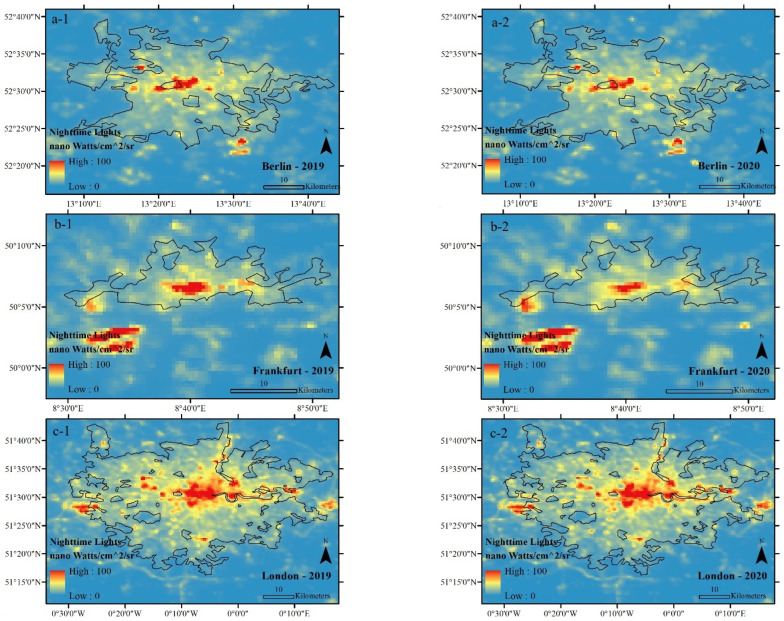
Average NLs of the studied cities during CSR, extracted from VIIRS in nano Watts/cm^2^/sr. (**a-1**) Average NLs of Berlin in 2019, (**a-2**) Average NLs of Berlin in 2020, (**b-1**) Average NLs of Frankfurt in 2019, (**b-2**) Average NLs of Frankfurt in 2020, (**c-1**) Average NLs of London in 2019, (**c-2**) Average NLs of London in 2020, (**d-1**) Average NLs of Madrid in 2019, (**d-2**) Average NLs of Madrid in 2020, (**e-1**) Average NLs of Paris in 2019, (**e-2**) Average NLs of Paris in 2020.

**Figure 12 ijerph-19-06579-f012:**
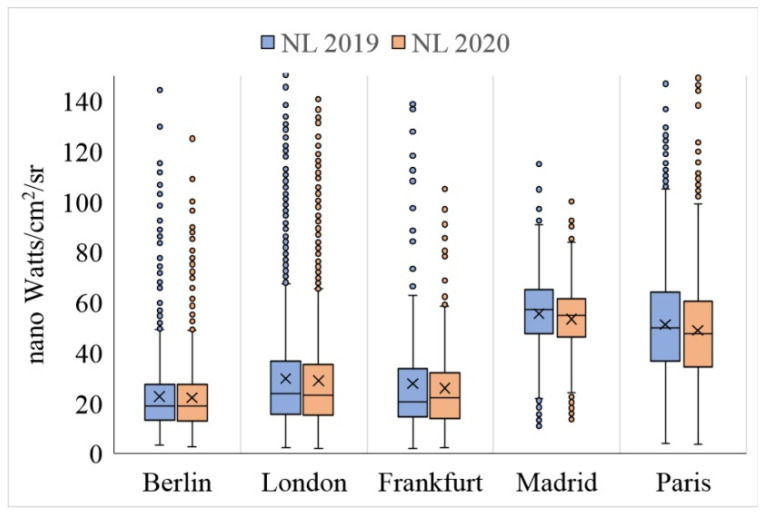
NL characteristics in the studied cities during 2019 (blue) and 2020 (red).

**Figure 13 ijerph-19-06579-f013:**
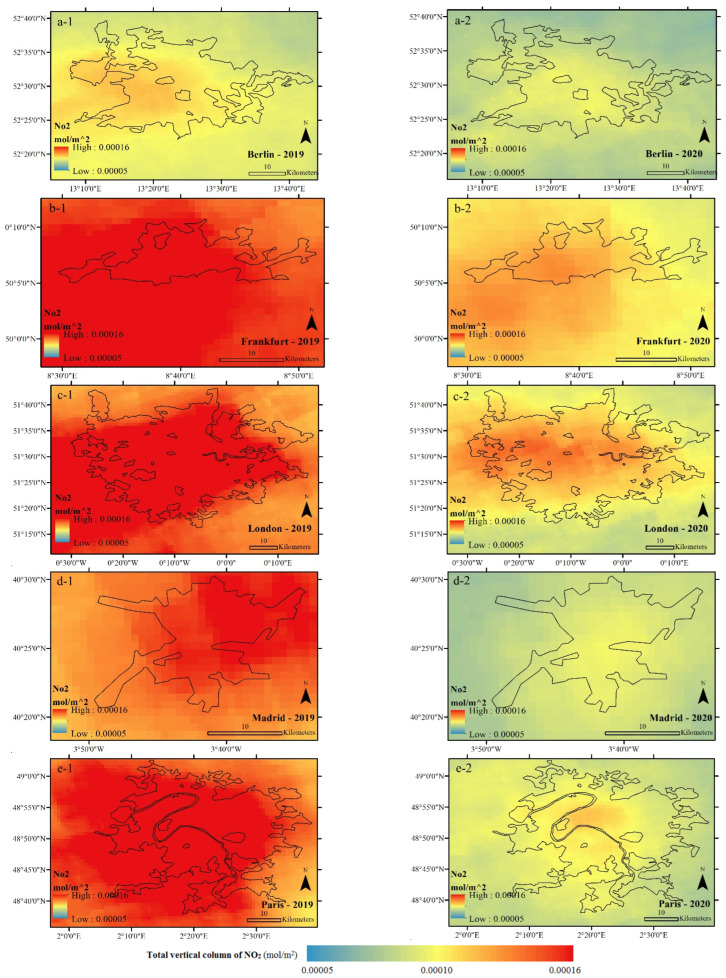
Average NO_2_ concentration of the studied cities during CSR, extracted from Sentinel 5P data in mol/m^2^. (**a-1**) Average NO_2_ concentration of Berlin in 2019, (**a-2**) Average NO_2_ concentration of Berlin in 2020, (**b-1**) Average NO_2_ concentration of Frankfurt in 2019, (**b-2**) Average NO_2_ concentration of Frankfurt in 2020, (**c-1**) Average NO_2_ concentration of London in 2019, (**c-2**) Average NO_2_ concentration of London in 2020, (**d-1**) Average NO_2_ concentration of Madrid in 2019, (**d-2**) Average NO_2_ concentration of Madrid in 2020, (**e-1**) Average NO_2_ concentration of Paris in 2019, (**e-2**) Average NO_2_ concentration of Paris in 2020.

**Figure 14 ijerph-19-06579-f014:**
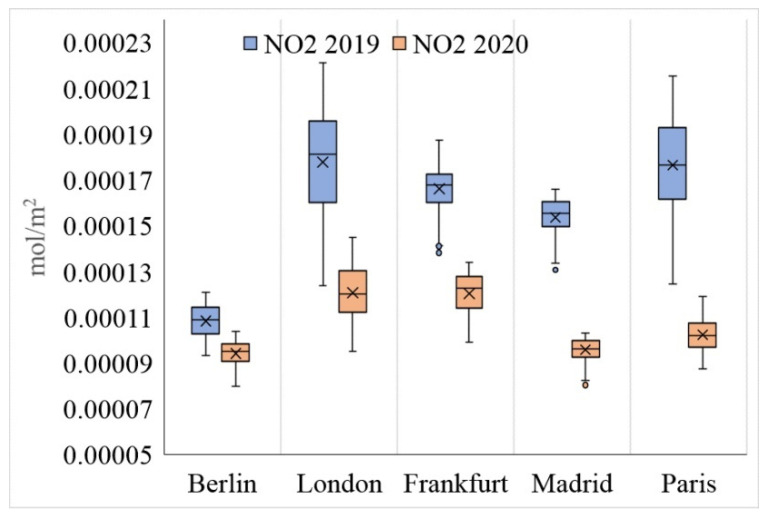
NO_2_ concentration in the studied cities during 2019 (blue) and 2020 (red).

**Figure 15 ijerph-19-06579-f015:**
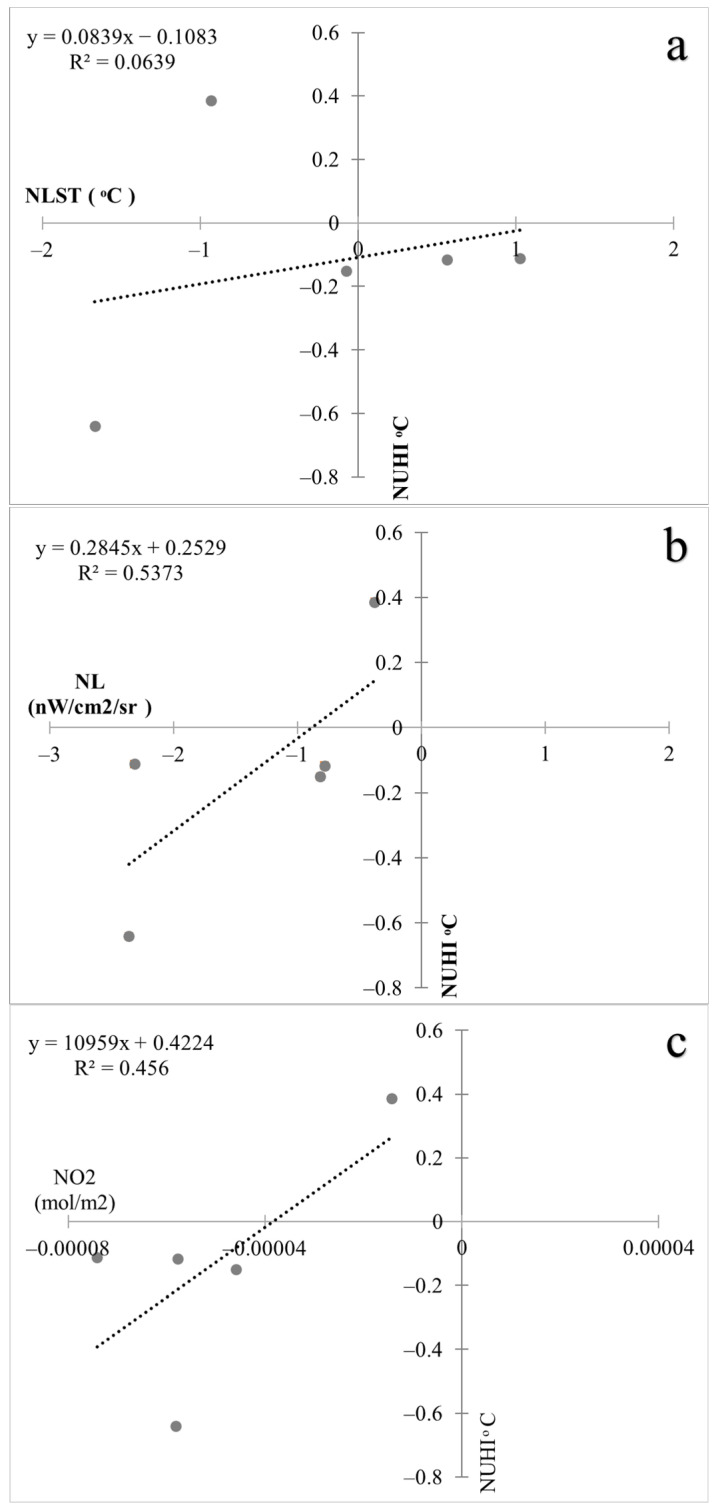
(**a**): relationship between NUHI changes and changes in NLST, (**b**): relationship between NUHI changes and changes in NL, (**c**): relationship between NUHI changes and changes in NO_2_ in the studied cities (2019–2020).

**Figure 16 ijerph-19-06579-f016:**
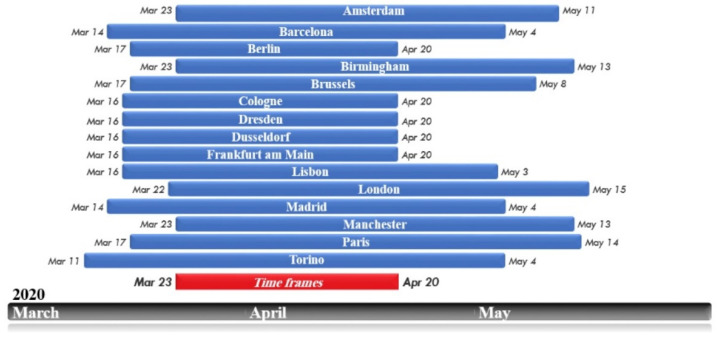
Time periods for executing severe restrictions on COVID−19 in European cities, and the time frame for conducting this study is shown in the red box.

**Figure 17 ijerph-19-06579-f017:**
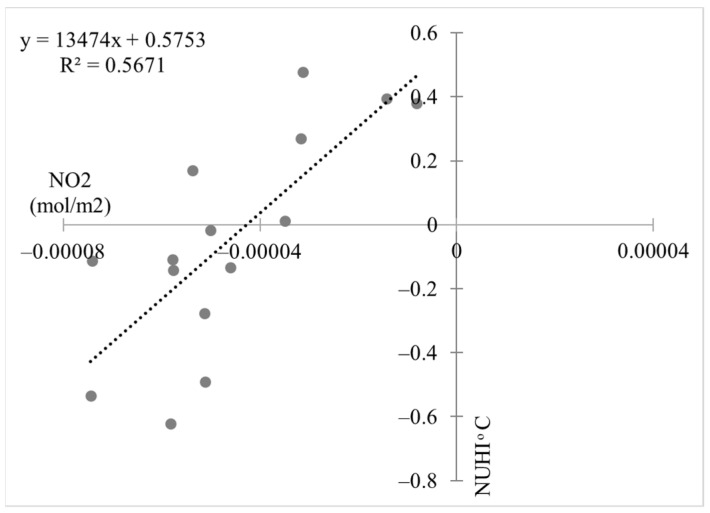
Relationship between NO_2_ and NUHI changes in European major cities.

**Table 1 ijerph-19-06579-t001:** Results of a paired-samples *t*-test for NLST data belonging to the studied cities for the two time periods.

Urban	Paired Differences					t	df	Sig. (2-Tailed)
	Mean	Std. Deviation	Std. Error Mean	95% Confidence Interval of the Difference				
				Lower	Upper			
Berlin	−0.93	0.52	0.02	−0.97	−0.90	−49.167	740	0.000
Frankfurt	−0.07	0.42	0.04	−0.14	0.00	−2.104	141	0.037
London	0.56	0.50	0.01	0.54	0.59	48.342	1842	0.000
Madrid	−1.67	0.67	0.05	−1.77	−1.57	−34.192	188	0.000
Paris	1.03	0.43	0.01	1.00	1.05	79.849	1127	0.000

**Table 2 ijerph-19-06579-t002:** Results of a paired-samples *t*-test for NUHI data belonging to the studied cities for the two time periods.

Urban	Paired Differences					t	df	Sig. (2-Tailed)
	Mean	Std. Deviation	Std. Error Mean	95% Confidence Interval of the Difference				
				Lower	Upper			
Berlin	0.39	0.55	0.02	0.35	0.43	19.226	740	0.000
Frankfurt	−0.15	0.46	0.04	−0.23	−0.07	−3.888	141	0.000
London	−0.12	0.45	0.01	−0.14	−0.10	−11.123	1842	0.000
Madrid	−0.64	0.66	0.05	−0.73	−0.55	−13.450	188	0.000
Paris	−0.11	0.45	0.01	−0.14	−0.09	−8.381	1127	0.000

**Table 3 ijerph-19-06579-t003:** Correlation between NUHI and NLST in the studied cities during 2019 and 2020.

Urban	NLST	
	2019	2020
Berlin	0.847 **	0.816 **
Frankfurt	0.772 **	0.843 **
London	0.885 **	0.922 **
Madrid	0.895 **	0.877 **
Paris	0.973 **	0.929 **

** Correlation is significant at the 0.01 level.

**Table 4 ijerph-19-06579-t004:** Areas of each NUHI class in the two periods of lockdown and the year before it (km^2^).

Urban	<2		>2	
	2019	2020	2019	2020
Berlin	281	206	460	535
Frankfurt	87	87	55	55
London	591	743	1252	1100
Madrid	14	58	175	131
Paris	203	246	925	882

**Table 5 ijerph-19-06579-t005:** Results of the paired samples *t*-test for NLs data in the studied cities for the two time periods.

Urban	Paired Differences					t	df	Sig. (2-Tailed)
	Mean	Std. Deviation	Std. Error Mean	95% Confidence Interval of the Difference				
				Lower	Upper			
Berlin	−0.38	2.72	0.10	−0.58	−0.18	−3.800	734	0.000
Frankfurt	−0.82	6.79	0.58	−1.97	0.33	−1.411	136	0.160
London	−0.78	3.31	0.08	−0.94	−0.63	−10.056	1808	0.000
Madrid	−2.37	5.75	0.42	−3.19	−1.54	−5.645	187	0.000
Paris	−2.32	4.67	0.14	−2.59	−2.04	−16.476	1099	0.000

**Table 6 ijerph-19-06579-t006:** Results of paired-samples *t*-test for NO_2_ concentration data in the studied cities for the two periods.

Urban	Paired Differences					t	df	Sig. (2-Tailed)
	Mean	Std. Deviation	Std. Error Mean	95% Confidence Interval of the Difference				
				Lower	Upper			
Berlin	−0.000014	0.000005	0.000000	−0.000015	−0.000014	−79.711	740	0.000
Frankfurt	−0.000046	0.000006	0.000000	−0.000047	−0.000045	−93.970	141	0.000
London	−0.000058	0.000018	0.000000	−0.000059	−0.000057	−140.434	1842	0.000
Madrid	−0.000058	0.000007	0.000001	−0.000059	−0.000057	−107.355	188	0.000
Paris	−0.000074	0.000014	0.000000	−0.000075	−0.000073	−177.655	1127	0.000

**Table 7 ijerph-19-06579-t007:** Difference in the averages of NUHI, NLST, NL, and NO_2_ during COVID-19 lockdown compared to its previous year.

Urban	UHI	LST	NL	NO_2_
Berlin	0.39	−0.93	−0.38	−0.000014
Frankfurt	−0.15	−0.07	−0.82	−0.000046
London	−0.12	0.56	−0.78	−0.000058
Madrid	−0.64	−1.67	−2.37	−0.000058
Paris	−0.11	1.03	−2.32	−0.000074

**Table 8 ijerph-19-06579-t008:** Differences in NUHI, NLST, NL, and NO_2_ changes during the lockdown compared to its previous year in European major cities.

Urban	NUHI	NLST	NL	NO
Amsterdam	0.48	−0.45	0.14	−0.000031
Barcelona	−0.54	−0.83	14.51	−0.000075
Berlin	0.39	−0.93	−0.44	−0.000014
Birmingham	−0.02	0.15	1.28	−0.000050
Bruxelles	0.17	0.88	−2.76	−0.000054
Dresden	0.38	−2.51	1.66	−0.000008
Dusseldorf	−0.49	−0.54	−0.90	−0.000051
Frankfurt Am Main	−0.13	−0.07	−1.80	−0.000046
Koln	−0.14	−0.05	−0.66	−0.000058
Lisboa	0.01	−1.13	−15.35	−0.000035
London	−0.11	0.57	−0.79	−0.000058
Madrid	−0.62	−1.67	−2.64	−0.000058
Manchester	0.27	1.13	−1.42	−0.000032
Paris	−0.11	1.03	−2.55	−0.000074
Torino	−0.28	1.57	−4.28	−0.000051

**Table 9 ijerph-19-06579-t009:** Correlation matrix: Correlation between NUHI, NLST, NL, and NO_2_ changes in each of the studied cities.

		NLST	NL	NO_2_
NUHI	Pearson Correlation	0.228	−0.215	0.701 **
	Sig. (2-tailed)	0.395	0.423	0.002
NLST	Pearson Correlation		−0.158	−0.335
	Sig. (2-tailed)		0.560	0.205
NL	Pearson Correlation			−0.255
	Sig. (2-tailed)			0.340

** Correlation is significant at the 0.01 level (2-tailed).

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
