# Peer review of "Effects of COVID-19 Restriction Policies on Urban Heat Islands in Some European Cities: Berlin, London, Paris, Madrid, and Frankfurt"

_ijerph, 2022, doi:10.3390/ijerph19116579_

Round 1
Reviewer 1 Report
The value of this research is its attempt to understand the impact of human activities on UHI. The weaknesses of the study is the selection of one of the variables and the time of the day to analyze such impact. The main sources of Nitrogen Dioxide Emissions include Cars, trucks, and buses followed by power plants, diesel-powered heavy construction equipment, and other movable engines.
It is obvious that these activities decrease at night. The main question that was supposed to be answered should have been why London and Paris had an increase.
The selection of the nighttime has to be clarified. It is not clear to me what time during the night the remote sensing data were collected. Was it early at night, midnight, or 4:00am? The time will have an impact on heat emitted from the above-mentioned sources and the impervious surfaces.
It is unfortunate that they did read a similar study by:
Restrepo, Carlos E. “Nitrogen Dioxide, Greenhouse Gas Emissions and Transportation in Urban Areas: Lessons From the Covid-19 Pandemic.” Frontiers in Environmental Science, vol. 9, June 2021, p. 689985. DOI.org (Crossref), https://doi.org/10.3389/fenvs.2021.689985
Additional remarks
Lines 30-37: The sixth IPCC report shows that the average global temperature has risen alarmingly around 1.09 °C (in the range between 0.95-1.2 °C) during the years 2011-2020 [1]. At the same time, the results of various studies show that compared to the general trend of global warming, cities, and most specifically, densely populated cities or metropolises have had a faster warming trend [2-4], so that urban areas have experienced 2-3 °C increase in temperature during the twentieth century [5,6]. This indicates that urban spaces are getting warm at a much higher speed under the influence of general atmospheric warming as well as the phenomenon of UHI.
My comments: Comparing 2011-2020 trend to a century trend can be misleading.
Figure 3. I see the symbol related to suburbs over a wide range of land use types. Is this correct? How do you define suburbs? I think there a difference between suburbs and rural areas.
In addition, why did you use two types of scales (one at the bottom left and another in the middle of the right side}?
Lines 169 and 173: ……Kiriging (Ordinary) Method.
My comments: Do you mean Kriging (Ordinary) Method?
Figure 4. My comments: What is A, B, and C? Please clarify.
Lines 181 to 187. My comments: The entire paragraph is not clear.

Author Response
Dear Reviewer
First and foremost, we kindly appreciate the reviewers’ time and helpful comments supporting us to lift up the quality of the draft. We have accommodated all the concerns and comments the reviewers provided.
Reviewer1:
- As the reviewers pointed out, given the conditions caused by the outbreak of the COVIID-19, the purpose of this article was to investigate the impact of anthropogenic factors on the formation and development of Urban Heat Islands. Thus, according to the literature, one of the most important indicators of the intensity of anthropogenic activities is the concentration of NO2 cited individually in similar studies. NL and NLST indices have been used in this research as well.
- This issue has been adequately unpacked in detail in the revised version of the article (lines 252- 262).
- A review of the theoretical foundations related to Urban Heat Islands shows that the greatest difference in temperature between the city and its surroundings (which is the maximum intensity of the Urban Heat Islands) often occurs three to five hours after the sunset (Runnalls & Oke, 2000; Stewart et al., 2021; Van Hove et al., 2015). Accordingly, in this study, the nighttime LST data (at 22:30 local time) has been used to calculate the NUHI. In the case of NO2 data, their daytime average has been examined. Therefore, changes in NO2 due to anthropogenic activities can be reflected in the intensity of the NUHI.
- The suggested article has been reviewed and reported in the revised version of the article. Reference number 54.
- We thank the reviewers for pointing this out. The text was incomplete in the first manuscript. It has been corrected in the revised version (Line 31-35)
- In the new version, the term “surrounding area” has been used instead of “suburb”. These areas have permeable surfaces (non-built-up) and are located at a distance of at least 3 km from the city.
- The ordinary Kriging method was meant. Revision has been made to the text.
- To clear up the ambiguity, Figures 4 and 5 have been removed and replaced with additional explanations in the text (Lines 174 -184).
- Figure 3 is corrected.
We are grateful again for your corrective feedback and valuable comments. Considering the changes made to the manuscript, we hope that the quality of the current manuscript has been upgraded and the standards of the Journal are met.
Sincerely Yours,
Authors
Reviewer 2 Report
The present paper uses COVID restrictions as natural experiment to show the impact of human actions on NUHI. This is a good idea, and the implementation, with data from 5 urban areas, is meaningful.
Changes, particularly in the description of methods and the language should be included. This will facilitate a better assessment since some things are not clear. There are also some suggestions for improvement.
- Language: Overall, general editing should be carried out.
- Methods: The description around “NUHI = NLST-NLSTr” is not clear. First, the two areas being compared are not described clearly. It seems that NLST refers to an urban raster cell and NLSTr to areas outside. You used the term “suburban”, but this might not be the best term, since you mean non built-up area. This is the definition of suburb and suburban, for instance, in Wikipedia: “A suburb(suburban or suburban area) is an area outside of a principal city of a metropolitan area, which may include commercial and mixed-use, but is primarily a residential area.”. You should use a term that is not associated with a built-up area. Maybe just surrounding area or non-built surrounding area. Fig. 4 is also not clear: what specific points are being use for each of the urban points shown and why? Only the closest? An average of the points in green? Also, Also, you say “less RMSE than other methods”, what other methods? This should be fully specified. And what RMSE? Also, why are values for NUHI shown for non-urban cells in figure 9, should not be NUHI=0 since the closest non-urban cell to a non-urban cell is itself?
- 5 is also not clear: what is being shown in each of the panels? There are some labels, but they are not clear. It is probably best if the legend specifies what panels (a) (b) and (c) are.
- Section 3.2 is also not clear on the precise dates for the two measurements, and whether only two dates are being used or some average. It reads as an average: l. 214, “Information 214 and analysis belonged to two main time periods, namely 2019 (before the onset of the 215 COVID-19 epidemic) and 2020”. If you mean two precise dates those should be clear. Later, in l. 244 you talk about “average NLST from March 23rd to April 20th in 2019 and 2020”. Why don’t you say so in the methods section? This is a method choice, not a result. How was the average calculated? What about day-to-day variance? How large was it?
- 169 and 172, : kiriging Should be kriging.
- The method being used with the paired t-test can be seen as a difference-in-difference estimator. It might be clearer to state in the method section that you are using a DiD estimator.
- While the conclusions state that “research results showed that the anthropo- 476
- genic factors in a set of daily activities, e.g., communication, as well as industrial, service 477 and administrative activities, have had a significant and serious impact on the formation 478 and pattern of the NUHI and NLST changes.”. However, in the of NUHI the correlation in Berlin has the opposite sign as in the rest of urban areas. While some mention is done of this fact, noting that it is an unexpected result (and precisely because of this), it should be clearer how this qualifies the conclusions.
Author Response
Dear Reviewer
First and foremost, we kindly appreciate the reviewers’ time and helpful comments supporting us to lift up the quality of the draft. We have accommodated all the concerns and comments the reviewers provided.
- It has been done.
- The explanations of the NUHI calculation have been revised and rewritten. (Lines 174 -184). In the new version, the term “surrounding area” has been used instead of “suburb”. These areas have permeable surfaces (non-built-up) and are located at a distance of at least 3 km from the city.
- In the new edition of the article, to clear the ambiguity, Figures 4 and 5 have been removed and replaced by additional and simpler explanations (Lines 174 -184).
- In the revised draft, in section 2, we have provided additional explanations based on the reviewer’s suggestion (124-128).
- Ordinary Kriging method was meant. Revision has been made to the text.
- Since in DID method, the behavior of a variable is studied in two periods and between the intervention and control groups, considering the conditions of this research (examining the changes in a variable in two different time periods), Paired sample t-test was used.
- This paragraph has been reviewed and revised in more detail in the conclusion section (lines 490-497).
We are grateful again for your corrective feedback and valuable comments. Considering the changes made to the manuscript, we hope that the quality of the current manuscript has been upgraded and the standards of the Journal are met.
Sincerely Yours,
Authors
Round 2
Reviewer 1 Report
It has been improved.